# Carbon Sequestration in Harvested Wood Products in Hungary an Estimation Based on the IPCC 2019 Refinement

Éva Király [1,*], Zoltán Börcsök [2], Zoltán Kocsis [3], Gábor Németh [3], András Polgár [4] and Attila Borovics [1]

1   Forest Research Institute, University of Sopron, Várkerület 30/A, H-9400 Sárvár, Hungary
2   Innovation Center, University of Sopron, Bajcsy-Zsilinszky E. Str. 4, H-9400 Sopron, Hungary
3   Faculty of Wood Engineering and Creative Industries, University of Sopron, Bajcsy-Zsilinszky E. Str. 4, H-9400 Sopron, Hungary
4   Faculty of Forestry, University of Sopron, Bajcsy-Zsilinszky E. Str. 4, H-9400 Sopron, Hungary
*   Correspondence: kiraly.eva.ilona@uni-sopron.hu

**Abstract:** As wood products in use store carbon and can contribute to reducing the concentration of atmospheric $CO_2$, the improved and enhanced use of wood products can be a successful measure in climate change mitigation. This study estimates the amount of carbon stored in the Hungarian harvested wood product (HWP) pool and the $CO_2$ emissions and removals of the pool. According to our results, the total carbon stock of the Hungarian HWP pool is continuously increasing. We estimated the total carbon stock of the HWP pool to be 17,306 kt C in the year 2020. Our results show that the HWP pool in Hungary is a carbon sink in most parts of the time series, with some years where it turns to a source of emissions. We carried out a simple projection up to 2070, assuming a constant inflow for the projected years that is equal to the average inflow of the last five historic years. This resulted in a decreasing trend in $CO_2$ removals, with removals already very close to zero in 2070. We concluded that in order to achieve significant future carbon sinks in the HWP pool technological improvements are needed, such as increasing the lifetime of the wood products and expanding the carbon storage capacity of wood products by reusing and recycling wood in a cascade system.

**Keywords:** HWP; wood products; climate change mitigation; forest; carbon storage; carbon stock; $CO_2$ removals

## 1. Introduction

The carbon transport from forest to wood products through harvesting and industrial processing creates a substantial carbon storage in the harvested wood product pool (HWP) [1]. Wood products in use store carbon and increasing this carbon storage can contribute to reducing the concentration of atmospheric carbon dioxide ($CO_2$) [2]. The efficiency of carbon storage is primarily affected by technological advancement [3]. The wood industry progression has obvious impacts on the efficiency of this carbon transport and on the total carbon stored in the HWP pools through the development of innovative products as well as the promotion of the utilization ratio of the roundwood and recycling rates [4].

According to the Sixth Assessment Report of the Intergovernmental Panel on Climate Change (IPCC), the improved and enhanced use of wood products can be a successful measure in climate change mitigation, and there is strong evidence at the product level that material substitution provides benefits on average for climate change [5]. Carbon storage in wood products can be increased through enhancing the inflow of products in use or effectively reducing the outflow of the products after use [5].

In the last decades, the potential contribution of HWPs in reducing greenhouse gas (GHG) emissions has been extensively investigated [6–10] and has become an important forestry-related issue in international climate negotiations [11]. Starting with the second commitment period of the Kyoto Protocol (KP), accounting for the net emissions and

removals from the HWP pool was obligatory for all parties to the KP. For the EU member states, accounting for the HWP pool continues to be obligatory in the era of the Paris Agreement. A new IPCC methodological guideline document called the 2019 Refinement to the 2006 IPCC Guidelines [12] (herein after referred to as the Refinement) was developed to support countries in making their accounting more transparent and precise. In our study, we use the new methodology of the Refinement.

In Hungary, the amount of carbon stored in the HWP pool was first estimated by Börcsök et al. [13,14]. They used the Tier 1 methodology of the 2006 IPCC Guidelines [15] and the data published on the FAOSTAT website. Rüter [16], Pilli at al. [17], and Brunet-Navarro et al. [2] estimated the annual amount of $CO_2$ sequestered in wood products and the carbon stock of the HWP pool for all EU member states, including Hungary. Király and Kottek [18] estimated the carbon stock and $CO_2$ emissions and removals of the HWP pool from the domestic harvest using the methodology of the KP Supplement [19]. In the Hungarian National Greenhouse Gas Inventory (GHGI), a modified version of their calculation is reported [20].

Forests have a unique role in actively removing $CO_2$ from the atmosphere. The European forest-based sector and its markets represent a key component in achieving climate neutrality by 2050, as laid out by the European Green Deal [21,22]. The production of long-lasting wood products can contribute to the reaching of the international climate goals [22,23]. The climate neutrality target relies on the compensation for residual emissions by absorption from the land use and forestry sector, which requires the sector's net sink to nearly double by 2050 [22]. To achieve the ambitious long-term EU objectives with the support of the forestry sector, foresters are to be called on to play a more active role in increasing carbon uptake and reducing emissions [24], and national policies should place greater emphasis on climate mitigation goals. To promote a more direct incentive system, the Circular Economy Action Plan [25] has anticipated a new regulatory framework for the certification of carbon removals by 2023. A carbon farming initiative was introduced by the Farm to Fork Strategy [26] and by the EU Forest Strategy [27]. The Hungarian National Forest Strategy [28] also puts its main focus on sustainable forest management, climate mitigation, and biodiversity conservation objectives.

The forest industry and firewood production are important branches of the Hungarian economy. Forests sequester nearly 10% of Hungary's total $CO_2$ emissions. Annually, the forests in the country accumulate about 1.2 million tons of carbon (4.5 million tons of $CO_2$ equivalent). Hungary's forest cover is 20.9%, or 2,064,000 hectares. The forests are composed of 90.5% deciduous tree species and are typically mixed forest communities [29]. More than 40% of the forests have a plantation-like composition of non-native tree species, most of which are the result of afforestation in recent decades. This afforestation has typically been carried out under unfavourable, degraded site conditions, which have been significantly modified by human activities. An example is the Duna-Tisza sand flats in the Great Plain, which have been artificially drained and have now become a semi-desert habitat. Here, only the introduced black locust (*Robinia pseudoacacia*) and pine species (*Pinus sylvestris* and *Pinus nigra*) could be used for successful afforestation.

The 1.1 million hectares of state-owned forest are managed by 21 state forestry companies. The nearly 32,000 private forest managers typically manage small, fragmented areas (average management size of around 17 hectares). The total growing stock volume of Hungarian forests was 403.99 million $m^3$ in 2021. In the same year, 7.5 million $m^3$ gross volume of wood was harvested of the 14 million $m^3$ annual increment, which means a 54% harvesting intensity (Table 1). It is worth noting that plantations of non-native black locust (1,420,985 $m^3$), hybrid poplars (1,305,907 $m^3$), and pines (predominantly *Pinus sylvestris* and *Pinus nigra*; 1,198,471 $m^3$) account for more than half of the annual wood production (3,925,363 $m^3$) [29].

**Table 1.** Total wood harvest in Hungary between 2016 and 2021 (thousand m$^3$) [21].

|  | **2016** | **2017** | **2018** | **2019** | **2020** | **2021** |
|---|---|---|---|---|---|---|
| Gross above-ground volume of harvested wood | 7338 | 7576 | 7767 | 7315 | 6580 | 7523 |
| Net above-ground volume of harvested wood, including: | 6176 | 6317 | 6481 | 6174 | 5533 | 6621 |
| industrial wood | 2950 | 2862 | 3038 | 2892 | 2457 | 3124 |
| Firewood | 3226 | 3454 | 3443 | 3282 | 3076 | 3497 |

In 2021, the area of afforestation increased significantly compared to the previous years, totalling 7238 hectares. Thirty-two percent of the afforestation was carried out with white oaks (*Quercus robur* and *Quercus petraea*), twenty-two percent with turkey oak (*Quercus cerris*) and other hardwood species, thirty-three percent with black locust (*Robinia pseudoacacia*), and thirteen percent with native and hybrid poplar and willow. The distribution of the country's forests by primary function is as follows: production 59.5%, recreation 1.0 %, and protection 39.5 % [29].

In order to enhance the contribution of the forest-based sector in combating climate change, it is of paramount importance to analyse the current level of carbon sequestration in the forests and forest-based sectors. The objective of our study was to quantify the carbon storage and $CO_2$ emissions and the removals of the Hungarian HWP pool and to project the future emission trends, assuming no change in the inflow rates.

## 2. Materials and Methods

### 2.1. Data Sources

To obtain a comprehensive picture of the carbon stored in the HWP pool in Hungary, we studied several data sources and selected the dataset deemed to be the most reliable. Due to the characteristics of the methodology, which intends to describe the whole lifespan of the wood products, the data on a longer period of the past were necessary.

We used the database of Király and Kottek [18] as a starting point, and we reviewed the data sources mentioned in their publication. Among the domestic sources, we used the works of Aladár Halász [30–32], as well as the Statistical Yearbooks and Pocket Books published by the Central Statistical Office [33–41]. We also used international databases, such as the TIMBER database of the United Nations Economic Commission for Europe (UNECE), and the data from the ForesSTAT forestry database of the FAO.

We revised the dataset published by Király and Kottek [18] and supplemented it with the EUROSTAT data for the production, import, and export of wood products. The data for the semi-finished wood products, which were originally taken from the National Data Collection Programme (OSAP), were replaced with more accurate data collected by the Central Statistical Office for the years 2005–2020.

According to our data sources, gross felling ranged between 5700 and 8100 m$^3$ in Hungarian forests in the period from 1980 to 2020. The amount of slash left at the logging site has a downward trend from around 20% in 1980 to around 16% recently (Figure 1).

Although the amount of net roundwood removed from forests seems not to have had any increasing or decreasing trend in the 1980–2020 period, the amount of industrial roundwood removed is decreasing with the rising energy prices and the increasing firewood demand (Figure 2). In 1980, 60% of the net roundwood removed was industrial wood; however, by 2020 this ratio had decreased to 44% (Figure 3).

Regarding the amount of imported roundwood, it considerably decreased after 1991, while the amount of exported roundwood has been increasing (Figure 4); this means that the amount of domestically harvested wood processed domestically has been increasing.

The production of semi-finished harvested wood products has an overall increasing trend. The years after the political regime change in Hungary (1990–1995) are an exception

as in those years both the industrial wood removal and the HWP production experienced a significant reduction due to the ongoing privatisation leading to a temporarily unclear land tenure situation and to no harvest on large areas of forest land.

There are remarkable changes in the distribution of product types in the studied period (Figure 5). In the 1960s, sawnwood production had the greatest share in the HWP production, and wood-based panels had a minor part, whereas the share of wood-based panels has now increased to 80%. The amount of produced paper and paperboard has been continuously increasing. These trends are caused by changes in production technologies in the wood industry coupled with changes on the demand side.

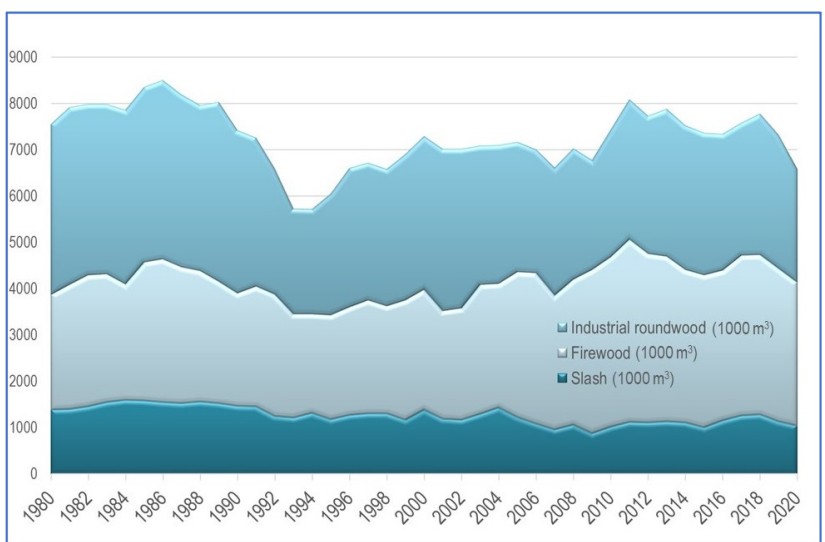

**Figure 1.** Amount of wood removal and slash in time period of 1980–2020.

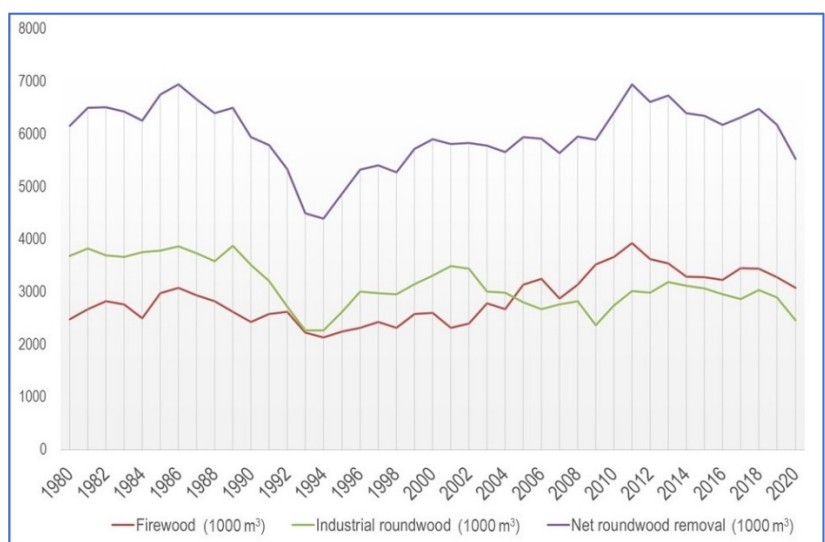

**Figure 2.** Amount of net roundwood, industrial wood, and firewood removed from forests in time period of 1980–2020.

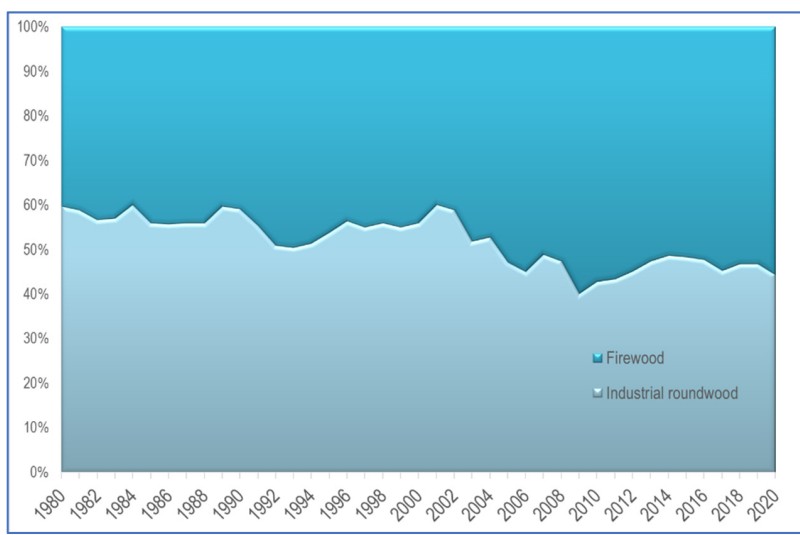

**Figure 3.** The proportion of industrial wood and firewood in the total net roundwood removal.

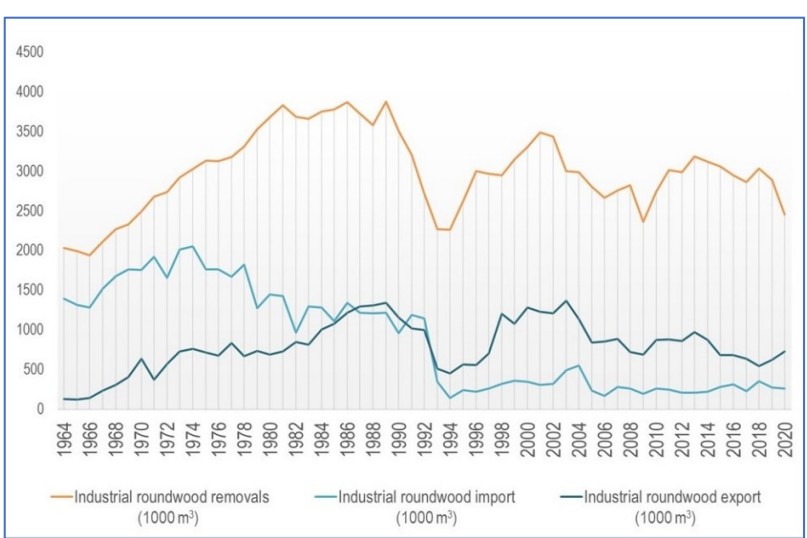

**Figure 4.** Industrial roundwood removal, import, and export in time period of 1964–2020.

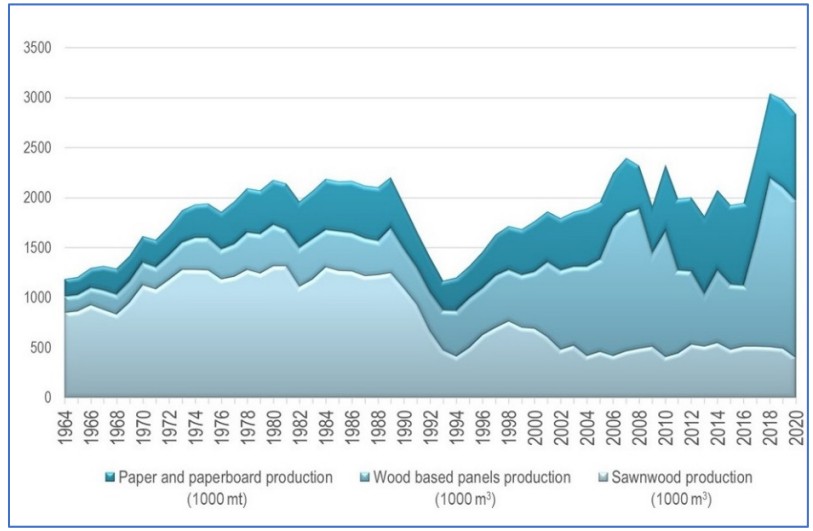

**Figure 5.** Production of semi-finished wood product types in time period of 1964–2020.

## 2.2. Methods of the Calculation

To estimate the HWP carbon stock changes, three different main approaches currently exist. These are the Atmospheric Flow Approach (AFA), the Production Approach (PA), and the Stock-Change Approach (SCA). These approaches are described in detail in the good practice guidance documents of the Intergovernmental Panel on Climate Change [12,15,19,42] and widely used for GHG reporting purposes. Using the PA, a country reports carbon stock changes associated with the HWPs from domestic harvests; so, while emissions from imports are excluded, those from exports are included [43]. This means that the PA may lead to the largest HWP carbon stocks for a country that has a large net HWP export [44,45]. In contrast, the SCA will result in the largest HWP carbon stocks for a country with a large net HWP import as the SCA estimates carbon stocks and emissions for all HWPs consumed by the reporting country, regardless of the country in which the harvest took place, i.e., imports are included and exports are excluded [10,46]. Finally, the AFA estimates the fluxes of carbon to and from the atmosphere, and all the HWPs consumed within the national boundary of the reporting country are considered, i.e., imports are included but exports are excluded. Consequently, for a country with a large net HWP export the use of the AFA will give reductions in its emissions [47].

These considerations suggest that HWP trade may be the main factor that affects the results of the use of the different approaches and that this significantly affects the HWP carbon stock and emission or removal profiles of a reporting country [10,44,46–48].

In view of the above, we decided to use a combination of the approaches recommended by the IPCC and estimated the carbon stocks of the domestically harvested HWPs as well as the HWPs from imported raw material. In order to do this, we used the PA methodology given in the Refinement, and we completed it with calculations for the HWP production from imported wood.

Table 2 shows the used half-life values and conversion factors which were taken from the Refinement.

**Table 2.** Default half-life values and conversion factors recommended by IPCC 2019 Refinement.

| | Half-Life (Year) | Density (Oven Dry Mass over Air Dry Volume) [Mg/m$^3$] | Carbon Fraction | C Conversion Factor (Per Air Dry Volume) [Mg C/m$^3$] |
|---|---|---|---|---|
| Coniferous sawnwood | 35 | 0.45 | 0.5 | 0.28 |
| Non-coniferous sawnwood | 35 | 0.56 | 0.5 | 0.225 |
| Veneer sheets | 25 | 0.505 | 0.5 | 0.253 |
| Plywood | 25 | 0.542 | 0.493 | 0.267 |
| Particle board | 25 | 0.596 | 0.451 | 0.269 |
| HDF | 25 | 0.788 | 0.425 | 0.335 |
| MDF | 25 | 0.691 | 0.427 | 0.295 |
| Fibreboard compressed | 25 | 0.739 | 0.426 | 0.315 |
| Insulating board (Other board, LDF) | 25 | 0.159 | 0.474 | 0.075 |
| | Half-Life (Year) | Relative Dry Mass (Oven Dry Mass over Air Dry Mass) [Mg/Mg] | | C Conversion Factor (per Air Dry Mass) [Mg C/Mg] |
| Paper and paperboard (aggregate) | 2 | 0.9 | - | 0.386 |

To estimate the magnitude of the carbon stock in the HWP pool in use and its net changes, the first-order decay function, as described in the Refinement, was used. The calculations were made separately for each product category. We assumed instantaneous oxidation at the end of the product's life cycle.

Concerning annual carbon stock change, Equation 12.2 of the Refinement was used:

$$\Delta C(i) = C(i+1) - C(i) \tag{1}$$

where

$$C(i+1) = e^{-k} \cdot C(i) + \left[ \frac{\left(1 - e^{-k}\right)}{k} \right] \cdot \text{inflow}(i) \tag{2}$$

$$C(1900) = 0,0 \tag{3}$$

*i: year; C(i): the carbon stock in the particular HWP commodity class i at the beginning of the year i, Mt C; k: decay constant of first-order decay for each HWP commodity class i given in units yr-1 (k = ln(2)/HL, where HL is the half-life of the particular HWP commodity in the HWP pool in years); inflow(i): the carbon inflow to the particular HWP commodity class i during the year i, Mt C yr-1; ΔC(i): carbon stock change of the HWP commodity class i during the year i, Mt C yr-1.*

As a proxy, it is assumed that the HWP pool is in a steady state at the initial time (1963) from which the activity data started, and $\Delta C(t0)$ is assumed to be equal to 0. This steady-state carbon stock $C(t0)$ for each HWP commodity class *i* is approximated based on the average of inflow (*i*) during the first 5 years (1964–1968), for which statistical data are available and are deemed reliable.

$$C_i(t0) = \frac{\text{inflow}_{i\ \text{average}}}{k} \tag{4}$$

In order to separate carbon in the imported HWPs, a feedstock factor $f_R(i)$ was used. The share of the particular feedstock commodity class *R*, originating from the domestic harvest in its total consumption for the semi-finished HWP production in the year *i*, was calculated.

$$f_R(i) = \frac{R_p(i) - R_{\text{ex}}(i)}{R_p(i) + R_{\text{imp}}(i) - R_{\text{ex}}(i)} \tag{5}$$

*fR(i): feedstock factor; R: industrial roundwood or wood pulp feedstock category; $R_p(i)$: production of the particular HWP feedstock commodity class in the year i, in $m^3$ or Mt; $R_{imp}(i)$: import of the particular HWP feedstock commodity class in the year i, in $m^3$ or Mt; $R_{ex}(i)$: export of the particular HWP feedstock commodity class in the year i, in $m^3$ or Mt.*

We estimated the $CO_2$ emissions and removals arising from the carbon stock change in the domestically harvested and consumed HWPs and from the carbon stock change in the domestically harvested and exported HWPs separately to increase transparency. Therefore, the annual carbon inflow to the HWP pool of the particular domestically consumed HWP commodity class (InflowPADC(*i*)) was calculated using the following equation.

$$\text{Inflow}_{\text{PADC}}(i) = [\text{HWP}_{\text{DPi}}(i) - \text{HWP}_{\text{EXi}}(i) \cdot f_{Ri}(i)] \cdot cf_i \tag{6}$$

*InflowPADC(i): annual carbon inflow to the HWP pool of the particular domestically consumed HWP commodity class; $HWP_{DPi}(i)$: production of the particular semi-finished HWP commodity class originating from domestic harvest in the year i, in $m^3$; $HWP_{EXi}(i)$: export of the particular semi-finished HWP commodity class originating from domestic harvest in the year i, in $m^3$; fR(i): feedstock factor; $cf_i$: carbon conversion factor of the particular semi-finished HWP commodity class i (see Table 1).*

With the above-described methodology, we estimated the carbon stock and net emissions/removals from the HWP pool separately for products from imported raw material, for products produced from domestically harvested raw material and consumed

domestically, and for products produced from domestically harvested raw material and subsequently exported.

To better understand the HWP pool dynamics and obtain a very simplified picture of the future tendencies of the emissions and removals from the HWP pool, we carried out a very simple projection. For the years between 2021 and 2070, we assumed that the inflow to the HWP product categories would be equal to the average inflow of the last five years (2016–2020). Due to the simple decay function applied, the outflow from the HWP pool is determined by the inflow of the previous decades. Thus, with this approach, we could test the impact of past determinations on future emissions assuming no future changes of the actual inflow rate.

## 3. Results

As shown in Figure 6, the total inflow to the HWP pool has an increasing trend, except for the years after the political regime change in 1990–1995. The share of the HWPs produced from domestically harvested wood has been increasing in the last decades.

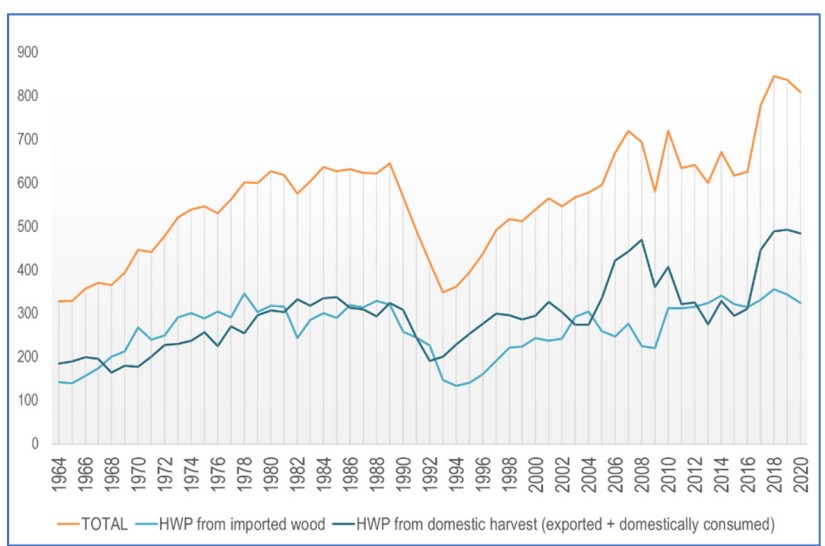

**Figure 6.** Inflow to the HWP pool from imported wood and from domestic harvest in time period of 1964–2020 (kt C/year).

The total inflow calculated for 2020 amounts to 810 kt C, and the inflow from the domestically harvested wood is equal to 485 kt C.

According to our results, the total carbon stock of the Hungarian HWP pool is continuously increasing, as is the carbon stock of the HWPs from the domestic harvest (Figure 7). However, the stock of HWPs from imported raw material is decreasing. The total carbon stock of the HWP pool amounts to 17,306 kt C, and the carbon stock of the HWPs from the domestic harvest amounted to 12,153 kt C in the year 2020.

Brunet-Navarro et al. [2] conducted a study in which they estimated the carbon stock in the European wood product sector (EU-28) and also at the member state level using the PA. According to their calculations, the Hungarian HWP carbon stock amounted to 10,002 kt C in 2015. For 2015, we calculated a stock of 11,381 kt C for the HWPs from the domestic harvest (i.e., as applied by the PA). The Hungarian GHGI reports 9075 kt C for 2015, using the PA and conservative estimates. This leads to the conclusion that the three estimates produced roughly comparable results, with a difference of around a thousand kt C in the carbon stock.

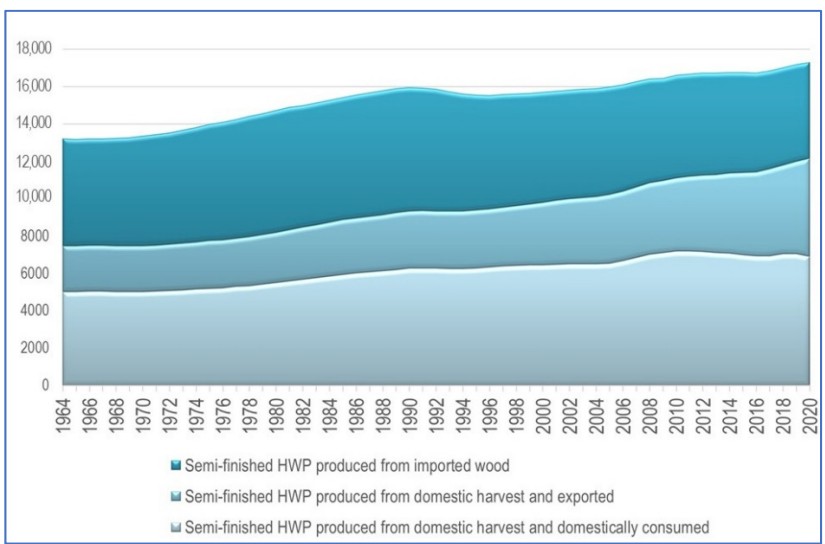

**Figure 7.** Carbon stock of the HWP pool 1964–2020 (kt C).

As shown in Figure 8, the HWP pool in Hungary is a carbon sink in most parts of the time series, with some years where it turns to a source of emissions, such as around the political regime change and in 2013–2016. The part of the HWP pool originating from the domestic harvest is producing bigger carbon removals due to the fact that in the past decades more HWPs were produced from imported raw materials than they are nowadays. This is also the reason why the HWP pool from the imported wood subcategory has been behaving as a source since 1989.

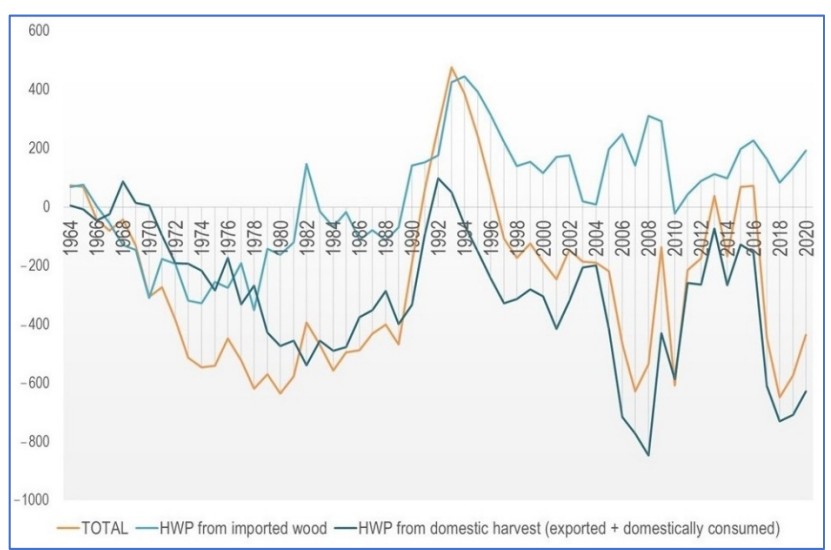

**Figure 8.** Emissions and removals from the HWP pool in time period of 1964–2020 (kt $CO_2$).

The predicted annual net removals from the HWP pool decrease from the historic value of $-436$ kt $CO_2$ in 2020 to the $-39$ kt $CO_2$ value predicted for 2070. This indicates that assuming a constant inflow equal to the average inflow of the last five historic years leads to a decreasing trend in the $CO_2$ removals. In the 2070s, the predicted removals are already very close to zero (Figure 9).

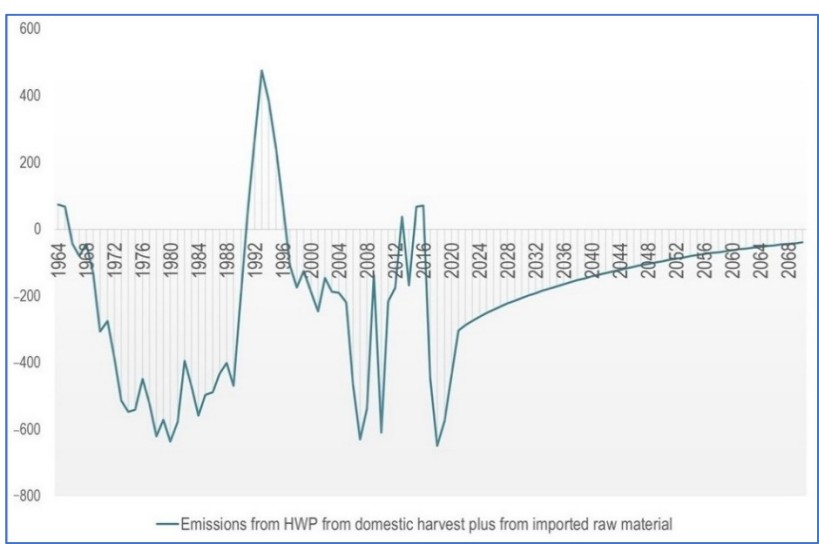

**Figure 9.** Historic and predicted emissions and removals from the HWP pool for time period of 1964–2070 (kt $CO_2$).

## 4. Discussion

Our results show that the Hungarian HWP pool is storing a significant amount of carbon, which has been continuously increasing since the 1960s. However, as the projected values show, without further action this trend may reverse in the upcoming decades. It is essential to increase the inflow or decrease the outflow in order to achieve continuous carbon sinks in the HWP pool.

Our estimate is somewhat different from the data reported in the Hungarian GHGI because in the GHGI a conservative estimate is made in order not to overestimate the removals or underestimate the emissions. The Hungarian GHGI currently uses the methodology of the KP Supplement [19], and only the HWP pool from the domestic harvest is accounted for. The production dataset underlying this study is also different from the data used for the calculation in the GHGI. Under the KP, the HWP emissions and removals were accounted for against a Forest Management Reference Level (FMRL). The contribution of HWPs to the Hungarian FMRL was estimated based on the historic HWP production data derived from the TIMBER database for the years 1964–2009 and the harvest quantities projected by the Model GLOBIOM up to 2020. The HWP emissions and removals were calculated using the C-HWP-Model, which estimates delayed emissions on the basis of the annual stock change of semi-finished wood products [16]. The average projected HWP emissions for 2013–2020 were incorporated into the FMRL. Following the due technical correction of the FMRL, as required by paragraphs 14 and 15 of the Annex to Decision 2/CMP.7, the average estimated HWP removal for 2013–2020 was −45 kt $CO_2$. Our estimate for the average removal of this period for domestically harvested wood products is −413 kt $CO_2$. The estimated carbon removal is larger than the one incorporated in the FMRL due to several effects, i.e., the difference in the production database, which has been updated; the difference between the actual harvest rates and the harvest projection; and the difference in the calculation methodology.

The base year for Hungary under the KP was the average of 1985–1987. According to the GHGI submitted in 2022, the net removals from the HWP pool in the base year amounted to −406 kt $CO_2$. Our estimate for the base year is very close to this value, amounting to −402 kt $CO_2$ for the HWPs originating from the domestic harvest. For 2020, we estimated the net removals to be −629 kt $CO_2$ from the domestically harvested part of the HWP pool. This means that the carbon sequestration of the pool has increased by fifty percent since the base year, which is in line with the increase in HWP production.

As a similar study for Lithuania states, regardless of the accounting method the main factors influencing the carbon stock in the HWP pool and its changes are domestic wood

supply and product half-life [49]. Brunet-Navarro et al. [2] point out how much the carbon stock in wood products could be increased by using higher proportions of harvested wood for those products with long lifespans and high recycling rates. According to their calculations, the carbon stock in wood products can be vastly increased by improving the cascade chains, with obvious climate change mitigation effects [2]. Even if we could only partially use this potential, it would lead to significant increase in carbon sequestrations. As a US study states, technological advancement serves as an important method to potentially increase the HWP carbon storage by as much as 44% [4]. As the availability of the woody biomass is insufficient to replace all the emission-intensive products and fossil energy, there is a need to focus on wood uses that provide the largest emission reductions [22]. It is important to increase the lifetime of wood products with product development and with the introduction of new smart technologies based on the typical characteristics of domestically produced timber [22]. Product longevity can be increased by changing the product portfolios or the end uses of the wood products, whereas shifts in wood uses may also lead to changes in the volume and allocation of the harvest to different assortments and tree species [22]. The carbon storage capacity of wood products can be supplemented and increased by reusing and recycling. Wood-based products should be reused and recycled in a cascade system, and energy recovery should be preferred over going to a landfill [22]. The development of recycling technologies should also be an objective for the future.

It is important to emphasize that the climate change mitigation potential of wood products is closely related to that of forests. As the main body of the terrestrial ecosystem, the forest is a carbon pool which has a strong function of carbon sequestration and storage. To fully understand the mitigation potential offered by forest-based climate change mitigating activities, a holistic approach is needed that considers forests and wood use options together for their overall contribution to the achievement of policy targets [22]. This approach must cover all the relevant carbon pools and fluxes of the forest ecosystems (in biomass, deadwood, and soil), wood products, and avoided emissions through material and energy substitution, as well as any leakage and rebound effects [22]. A trade-off between increased wood product carbon stock and decreased forest carbon sinks might exist. However, these trade-offs are often time-dependent and activities that provide net mitigation benefits in the short-term may limit or complicate climate change mitigation in the long-term. For example, maximizing carbon storage in forest ecosystems through forest conservation or reduced harvest may help to achieve mitigation targets in the short-term, but this benefit will become smaller as the forests grow older and their growth rate reduces [22]. Climate change-induced disturbances can also negatively impact on forest growth or result in significant mortality and thus lead to carbon losses [50]. While forests play an important role in climate change mitigation, they are also affected by climate change and require adaptation. Tree species distribution modelling shows that, in a wide range of climate scenarios, the second half of this century will see almost all the main European tree species experience reductions in their suitable areas, especially in eastern and southern Europe [51].

High temperatures and high concentrations of carbon dioxide are widely believed to stimulate tree growth, forest productivity, and carbon uptake. However, climate change may also increase the carbon turnover through increases in respiration, background mortality, and disturbances. As a result, additional carbon is removed from the atmosphere, but it is also being released back more swiftly [52]. Increasing disturbances and increasing productivity under climate change are linked. Forest productivity is affected by tree species selection. Forest management can actively shape the species composition of forests, favour the species mixtures that make forest resilient to changes, and support species range shifts by planting or naturally regenerating species that are thought to cope better with changing local conditions [22]. However, species range shifts would likely entail substantial changes to carbon sequestration as there is a trade-off between growth and longevity for almost all tree species. Currently, young stands and fast-growing species are making a substantial contribution to the carbon sink, but the forests dominated by these species face increasing disturbance risks and are increasingly being converted into mixed stands. If these species

are mixed with species that are less productive in terms of carbon capture, the productivity and carbon sequestration potential may decrease [53], which may lead to changes in the harvest rates and available wood assortments, thereby having a significant impact on the wood industry as well.

## 5. Conclusions

In our study, we estimated the carbon stock and the stock change of the Hungarian HWP pool using a combination of approaches recommended by the IPCC. We separately estimated the stock changes of the HWPs produced from imported wood and the stock changes of the HWPs from the domestically consumed and exported domestic harvest. Based on our estimate, we can conclude that the PA is the most favourable HWP accounting approach for Hungary as the part of the HWP pool originating from the domestic harvest is producing the biggest carbon removals. In contrast, the HWP pool from the imported wood subcategory has been behaving as a source of emissions since 1989. This means that omitting imported wood from the carbon accounting leads to the largest net removals. The PA is also in line with the methodology of the Refinement and is consistent with the practice of most of the EU member states. It is therefore advisable for Hungary to continue to apply the PA in its GHGI.

We estimated larger net removals as compared to the Hungarian GHGI due to the update in the underlying production database. For the years 2005–2020, our dataset contains the comprehensive data of the Hungarian Central Statistical Office, which is deemed to be the most reliable data source currently available. Therefore, a recalculation of the GHGI emission time series should be considered. However, the requirement of conservativeness in GHG reporting is also essential and should be met when using the new dataset. It is also worth considering switching to the new methodology of the Refinement as it is more transparent regarding the separate presentation of the emissions arising from the change in the carbon stock in domestically consumed HWPs and the change in the carbon stock in exported HWPs.

Assuming a constant inflow rate, the future $CO_2$ removals of the Hungarian HWP pool decrease, as our simple projection shows. In the 2070s, the predicted removals are already very close to zero. This means that to achieve continuous or increasing carbon sinks in the HWP pool it is essential to increase the inflow to the pool or decrease the outflow from the pool. Appropriate policy decisions in line with the latest research findings on mitigation and adaptation are needed to successfully combat climate change. European forests and wood products can provide a significant contribution to the achievement of climate neutrality. Mitigation activities, such as avoiding deforestation, afforestation/reforestation, shifts in wood use, cascading, and increased efficiency, should be applied in combination [22]. The types of wood use that give the largest net emission reductions should be prioritized as the availability of the woody biomass is and will be vastly insufficient to replace all the emission-intensive products and fossil energy [22]. To achieve the climate policy targets, carbon sequestration and storage in ecosystems and in wood products is an essential tool of high importance. However, the reduction in and strict control of carbon emissions is also inevitable.

In the framework of our ForestLab project (TKP2021-NKTA-43), we are planning to develop a new model covering the life cycle of Hungarian wood products from the production to reuse, recycling, and waste management. This would allow the prediction of the future impacts of changing half-lives, recycling rates, and waste management practices on the net emissions or removals of the HWP pool. In so doing, we would get an idea of the climate mitigation potential of the wood production and recycling technology changes and that of wood waste management, thereby supporting political decision-making.

**Author Contributions:** Conceptualization, É.K. and A.B.; methodology, É.K.; validation, Z.B., Z.K., G.N., A.P. and A.B.; formal analysis, É.K.; investigation, É.K., Z.B., Z.K., G.N., A.P. and A.B.; data curation, É.K.; writing—original draft preparation, É.K. and A.B.; writing—review and editing, Z.B., Z.K., G.N., A.P. and A.B.; visualization, É.K.; supervision, A.B.; project administration, A.B.; funding acquisition, A.B. All authors have read and agreed to the published version of the manuscript.

**Funding:** This article was made in frame of the project TKP2021-NKTA-43, which has been implemented with the support provided by the Ministry of Innovation and Technology of Hungary (successor: Ministry of Culture and Innovation of Hungary) from the National Research, Development and Innovation Fund, financed under the TKP2021-NKTA funding scheme.

**Institutional Review Board Statement:** Not applicable.

**Informed Consent Statement:** Not applicable.

**Data Availability Statement:** All data that are necessary for the reconstruction of this study can be found in this paper and in the referenced sources.

**Conflicts of Interest:** The authors declare no conflict of interest.

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
