# Peer review of "Carbon Sequestration in Harvested Wood Products in Hungary an Estimation Based on the IPCC 2019 Refinement"

_forests, doi:10.3390/f13111809_

Round 1

Reviewer 1 Report

I am really inspired and encouraged by this work for its novelty and practicability for today's scenario. A really interesting work with excellent information. I found no technical problems. However, i have mentioned two comments embedded as sticky not in the manuscript also. Please address that. Overall, a good information for international readers. Please see that the authors have addressed my two comments. No need to send me again, if the editor is satisfied, the manuscript may please be processed further positively.

Author Response

Thank you very much for your comments.

Please find enclosed our detailed answers in a separate pdf doument.

Reviewer 2 Report

Kiraly et al. tried to study the carbon sequestration in harvested wood products in Hungary, which on its first look seems a good study. However, after thoroughly reading the manuscript, it seems that the authors have not justified with the study. Here are some of the flaws which I have seen in the study

Abstract is not well written. Sentences are without punctuations, so difficult to understand.

Introduction has been divided into unnecessary paragraphs.

The authors must understand that it is mandatory that all scientific names used must be italicized.

There is no justification of the study in the introduction section.  There are no defined objectives and hypothesis.

Materials and methods section is poorly written. In the first part of this section, most of the sentences which would have been in the results section were included.

Results section is again poorly written and presented. Even, one can find discussion in this section.

Discussion is to some extent well written but here the authors have failed to discuss their main finding with the earlier studies, thoroughly.

In these kind of studies, conclusion and recommendations are an important section which are completely absent in this study.

There is no outcome of this study and can’t be replicated.

Keep these flaws in consideration, I can’t endorse the manuscript for publication in the current form and I think these flaws can’t be corrected by a revision. So, I can’t endorse the paper for revision and I reject the manuscript.

Author Response

Thank you very much for your comments.

Please find enclosed our detailed answers in a separate pdf document.

Reviewer 3 Report

Dear Authors, 

I have read the document. I found it interesting how the policies of a country can change the balance of the budget as they did in Hungary from 1990 to 1995 for example. However, I would like to read more about this aspect. I would like to know that the policies of a country could be much more important in a country to combat climate change and not only the studies of carbon quantification in different forestry and agroforestry systems.

wood product (HWP) pool, Fix =  wood product pool (HWP) line 29

from line 85 to line 90. I would like to read the objective of the work, in order to better close the introduction of the work.

Author Response

(The authors gave the same response as above.)

Reviewer 4 Report

This article is very interesting and in line with the current research focus.it estimates the amount of carbon stored in Hungarian har- 13 vested wood product (HWP) pool and the CO2 emissions and removals of the pool. But I think the following aspects need to be improved or expanded.

1.As the main body of terrestrial ecosystem, forest is a carbon pool, which has a strong function of carbon sequestration and sink.High temperatures and high concentrations of carbon dioxide are widely believed to be "catalysts" that stimulate tree growth, helping trees mature faster and absorb more carbon.The carbon storage role of forests may be "ephemeral", and the article should pay due attention to and add content and consideration to this aspect.

2.There is a trade-off between growth and longevity for almost all tree species. Because slow-growing and persistent trees are increasingly replaced by fast-growing but fragile trees, resulting in a decline in CO2 absorption rate, the weight of which needs to be taken into account in the simulation. The article does some work, but is not comprehensive enough, such as the sharp increase in the chance of death as trees reach their maximum growth size?

The perspective of the article is relatively new, the forest will not necessarily change its role as a carbon sink, can not rely too much on the growth of unit area to maintain and enhance the potential of forest as a carbon sink, but should be achieved through sustainable ways to slow down deforestation and increase the size of forests. More importantly, it is necessary to strictly control carbon emissions. I hope my suggestions can give a hint to the quality of the article.

Author Response

(The authors gave the same response as above.)

Round 2

Reviewer 2 Report

I endorse the manuscript for publication 

Reviewer 3 Report

I was reading the whole document and now is much better. thank you for taking my suggestion.